# Comparison Studies on Several Ligands Used in Determination of Cd(II) in Rice by Flame Atomic Absorption Spectrometry after Ultrasound-Assisted Dispersive Liquid–Liquid Microextraction

**DOI:** 10.3390/molecules27030590

**Published:** 2022-01-18

**Authors:** Qian Sun, Xinyu Cui, Yanfeng Wang, Pingping Zhang, Wenjuan Lu

**Affiliations:** Institute of Materia Medica, Shandong First Medical University & Shandong Academy of Medical Sciences, Jinan 250062, China; sunqian201907@163.com (Q.S.); 18366034406@163.com (X.C.); wangyanfeng@sdfmu.edu.cn (Y.W.); zhangpingping@sdfmu.edu.cn (P.Z.)

**Keywords:** ligands, rice, cadmium, dispersive liquid-liquid microextraction, flame atomic absorption spectrometry

## Abstract

Ligands plays an important role in the extraction procedures for the determination of cadmium in rice samples by using flame atomic absorption spectrometry (FAAS). In the present study, comparative evaluation of 10 commercially available ligands for formation of Cd(II)-ligand complex and determination of cadmium in rice samples by ultrasound-assisted dispersive liquid–liquid microextraction (UADLLME) combined with FAAS was developed. Sodium diethyldithiocarbamate (DDTC) provided a high distribution coefficient as well as a good absorbance signal, therefore DDTC was used as a ligand in UADLLME. A low density and less toxic solvent, 1-heptanol, was used as the extraction solvent and ethanol was used as the disperser solvent. In addition, the experimental conditions of UADLLME were optimized in standard solution first and then applied in rice, such as the type and volume of extractant and dispersant, pH, extraction time, and temperature. Under the optimal experimental conditions, the detection limit (3σ) was 0.69 μg/L for Cd(II). The proposed method was applied for the determination of Cd(II) in three different rice samples (polished rice, brown rice, and glutinous rice), the recovery test was carried out, and the results ranged between 96.7 to 113.6%. The proposed method has the advantages of simplicity, low cost, and accurate and was successfully applied to analyze Cd(II) in rice.

## 1. Introduction

Cadmium has been widely used in electroplating, dyeing, plastics, alloys, batteries, rubber, and other industries. With the increasing production, a considerable amount of cadmium is discharged into the environment. Cadmium is highly harmful to plants, animals, and human health [1]. Additionally, it is considered to be one of the largest contaminants in phytogenic food [2]. Long-term accumulation of cadmium in the body causes damage to kidney, lung, and liver, as well as osteoporosis, hypertension, and even cancer [3,4,5]. The toxicity of cadmium in rice has been a worldwide concern. As the main grain in many countries, the quality of rice is directly related to human health [6,7]. Due to the special gene of rice, its roots have a stronger ability to absorb cadmium than other crops such as corn and soybean. Therefore, it is crucial to monitor the content of cadmium in rice [6,7,8,9].

At present, the trace amounts of cadmium is mainly analyzed by flame atomic absorption spectrometry (FAAS) [2,10,11,12,13,14,15,16,17,18], graphite furnace atomic absorption spectrometry (GFAAS) [19,20,21,22,23], inductively coupled plasma atomic emission spectrometry (ICP–OES) [24], and inductively coupled plasma mass spectrometry (ICP-MS) [25,26,27,28]. Compared with costly analysis technique, such as ICP-OES and ICPMS, FAAS is a conventional technique that has been widely applied in metals analysis because it is a less expensive and simpler alternative technique [29,30]. In addition, FAAS is faster, easier to operate, and has considerable accuracy and sensitivity. Laboratories are unlikely to make the switch to ICP-MS or ICP-OES, since the analysis can be performed using the existing FAAS methods, especially when the laboratory focuses on single element analysis, such as that for Cd(II) in rice. However, the determination of cadmium at the trace level, especially in rice samples that have complex matrices, is difficult when using FAAS without pretreatment procedures [13].

Dispersive liquid–liquid microextraction (DLLME) is an attractive method that has been used widely in the field of sample preparation since its innovation in 2006 [31,32]. The conventional and modified DLLME methods have been successfully applied to determine the trace amounts of cadmium in various matrices [15,16,17,18,19,22,23,24,25,26,27]. Generally, the pre-concentration of cadmium ions requires suitable complexing agents to form metal complexes and obtain high partition coefficients of cadmium between the aqueous phase and organic extraction solvents. The use of complexing agents that selectively react with cadmium and subsequent extraction of the obtained metal complex has been widely applied in analysis. For instance, cadmium has been complexed with 8-hydroxyquinoline (8-HQ) [2], 1-(2-thiazolylazo)-p cresol (TAC) [10], ammonium pyrrolidine dithiocarbamate(APDC) [12,20,22,24,25], 1-(2-pyridylazo)-2-naphthol (PAN) [13,27], 4,4-dimethyl2,2-bipyridine [15], diphenylcarbazone [16], sodium diethyldithiocarbamate (DDTC) [17,23,25], 1,3-dihydroxybenzene [18], L-Cysteine [19], tricaprylmethylammonium chloride(Aliquat^®^ 336) [26], sodium O,O-diethyl dithiophosphate (DEDTP) [33], N,N′-bis(salicylidene)ethylenediamine (BSE) [34], the obtained cadmium complexes extracted by ultrasound-assisted emulsification microextraction with the solidification of floating organic droplet [10], supramolecular solvent based liquid–liquid microextraction [12], solidified floating organic drop microextraction [13], vortex-assisted room temperature ionic liquid dispersive liquid–liquid microextraction [12,18], green solvent-based ultrasonic assisted dispersive liquid–liquid microextraction [19], switchable hydrophilicity solvent based liquid phase microextraction [20], ultrasound-assisted magnetic retrieval-linked ionic liquid dispersive liquid-liquid microextraction [22], lab-in-syringe magnetic stirring-assisted dispersive liquid-liquid microextraction [24], liquid phase microextraction [28], co-precipitation method [33], magnetic solid-phase extraction [34], and DLLME [2,15,16,17,23,25,26,27]. 

The presence of an appropriate complexing agent for extraction of Cd(II) is necessary in the mentioned DLLMEs [2,12,15,16,17,18,19,22,23,24,25,26,27], however, there are rare discussions about the effect of different kinds of ligand in DLLMEs. Most previous researchers demonstrated that the types of extraction solvent and disperser solvent were important in the DLLMEs technique. In fact, the organic ligands play an important role in the determination of trace Cd(II) because they are affected not only by the extraction efficiency of Cd(II) but also the FAAS behavior of metals. For example, the extraction conditions of Cd(II) when using DDTC and O,O-diethyl phosphordithiooic acid ammonium(DDTP) are quite different. Acidic conditions are suggested to enhance the extraction efficiency of DDTP, while DDTC is decomposed into diethylamine and carbon disulfide in an acidic solution [35]. Even DDTC, APDC, and sodium dimethyldithiocarbamate (DMDTC) have similar chemical structures, and the stability of the complexes with Cd(II) that will affect the FAAS procedure are different due to their different substituents. Hence, it is important to perform a comparative study in order to choose appropriate an ligand by consideration of the extraction efficiency of Cd(II) as well as atomization degree of the Cd(II)-ligand complex.

In our work, ultrasound-assisted dispersive liquid–liquid microextraction (UADLLME) was developed to deter–mine the trace amounts of cadmium in rice. Compared with the typical DLLME procedures, employing an ultrasound approach to this method led to some advantages, which can be summarized as follows: decrease of disperser solvent usage, the alternating acoustic pressure speeds up the extraction procedure, and increase of extraction efficiency [36]. Meanwhile, all experimental variables such as dispersant type and volume, ultrasonication time, extraction temperature, pH, coexisting ion interference, and other parameters were optimized. A procedure combining DDTC as the ligand for determination of trace cadmium in rice samples was established.

## 2. Results

### 2.1. Comparptive Studies

Of the different ligands that have been reported to have been used for determination of Cd(II), the comparative tests were made to investigate the effect of 10 ligands on the extraction and atomization degree of Cd(II). The structures of the 10 ligands were listed in Figure 1. The distribution coefficient (D) of Cd(II) between water and 1-heptanol was defined as the concentration of Cd(II) in the organic (cor) and aqueous phases (caq), and were calculated in the following Equation (1). The results are shown in Figure 2.
(1)$$D=\frac{c_{\mathrm{or}}}{c_{\mathrm{aq}}}.$$

From Figure 2, it can be seen that the ligands containing sulfur atoms (DDTC, APDC, DMDTC, DBDTC, and DDTP) have higher extraction efficiency than the five N-donor ligands (PAN, BSE, NTA, EDTA and DTPA) under the current experimental conditions. For the five N-donor ligands, the strength and content of hydrophilic groups have significant impacts on the extraction behavior of cadmium. The distribution coefficient was decreased by increasing the hydrophilic group strength and content. For the five S-donor ligands, the higher D values of S-donor ligands were directly related to two donor sulfur atoms. According to the hard-soft acids-bases (HSAB) principle, Cd(II) is a reasonably soft metal ion, which prefers to coordinate with ligands containing N-donor atoms or S-donor groups. Meanwhile, according to HSAB, the sulfur atom is classified as a softer base than the nitrogen atom and thus has a stronger interaction with the soft acid Cd(II) ions. Therefore, the ligands containing sulfur atoms have higher D of Cd(II).

However, highly efficient extraction is important but not necessary in the FAAS process. For example, DMDTC provided the highest D in the 10 studied ligands, but the absorbance signal was low. In contrast with DMDTC, although DDTC provides D values less than APDC and DMDTC, the absorbance was the highest in the 10 ligands. The reason is that organic complexing reagents exhibit various effects in FAAS.

In the FAAS process, the mechanisms by which ligands affect the FAAS behavior of metals are rather complicated. The Cd(II)-DDTC chelates were extracted into a nonaqueous solvent and were then nebulized into flames. In flames, the very fine aerosol droplets of the nebulized 1-heptanol solution containing Cd(II)-DDTC were desolvated and the resulting solid particles were thermally decomposed. The absorbance signal of cadmium is controlled by the number of free cadmium atoms in the flame. The free cadmium atoms appear by dissociation or reduction of the cadmium-chelate presents in the gaseous state after volatilization. The number of free cadmium atoms could be increased if volatile compounds are formed. However, a highly volatile complex, which is generated even at low temperatures, would yield low sensitivity and poor precision. Hence, Cd(II)-DDTC chelation is better thermally decomposed at the actual flame temperature.

The S-donor ligands containing different alkyl or pyrrolidine substituent groups have been investigated by taking consideration of steric and electronic effects. Obviously, the four alkyldithiocarbamates (R_2_N)C(S)S- have much higher D than alkylthiophosphate (RO)_2_P(S)S- (DDTP), which can be explained on the basis of the electronic property of the substituent groups. The electron-withdrawing groups such as phosphoryl group would result in lower electronic density on the S atom and hence lead to weaker protonation than those with electron-donating groups such as alkyl or pyrrolidine. In addition, the different alkyl (R) group in the dithiocarbamate ligand can affect the extraction ability of Cd(II). The D values of all of the four alkyldithiocarbamates (R_2_N)C(S)S- was increased by the stronger ability of donor-giving electrons.

The experimental results in Figure 2 show that the alkyl (R) group in the dithiocarbamates ligands can have significant impacts on the formation of thermally stable species. Among the four alkyl-dithiocarbamate compounds, i.e., DDTC, APDC, DMDTC, and DBDTC, only Cd(II)-DDTC exhibited high volatility and a high degree of atomization of cadmium in acetylene flame and produced a very good absorbance signal of cadmium. This is maybe because DMDTC and APDC formed stronger and more thermally stable cadmium-chelate than DDTC. According to previous inference [33,37], the volatile species of Cd-DDTC complexes in the organic solution was likely to be 1:2 metal-ligand complex, Cd[S_2_CN(C_2_H_5_)_2_]_2_. Unfortunately, we failed in our attempts to confirm the volatile species of Cd(II)-DDTC complexes and the mechanism for the thermal decomposition Cd(II)-DDTC complex in the acetylene-air flame. Our future work will focus on this.

### 2.2. UADLLME Procedure Optimization

In order to obtain the high extraction efficiency and absorbance signal of the proposed method, DDTC was chosen as the ligand to form complexes with the Cd(II) ions. Several parameters that influenced the extraction efficiency, including type and volume of extraction solvent and disperser solvent, pH, ultrasound extraction time, and temperature, were also investigated and optimized. The optimization experiments were all carried out three times and the average value was used.

#### 2.2.1. Effect of Extraction Solvent

The appropriate organic extraction solvent used in UADLLME is an important parameter that affects the extraction efficiency of Cd(II). It should be immiscible with water, have the ability to extract the Cd(II)-DDTC complex, and easily burn in FAAS. In previous studies, chlorinated solvents (e.g., chloroform and carbon tetrachloride) in the traditional DLLME method were frequently chosen as the extraction solvent. However, all of those solvents have high toxicity [27]. Therefore, we used the less toxic solvent 1-heptanol and 1-undecanol instead of chlorinated solvents usually used in DLLME.

Three types of organic solvents were studied: carbon tetrachloride, 1-heptanol, and 1-undecanol. The results in Figure 3a indicate that 1-heptanol and 1-undecanol have higher Cd(II) absorption than carbon tetrachloride. Carbon tetrachloride has the lowest absorbance signal, because in the presence of chlorinated extraction solvent, volatile chlorides of Cd(II) can be formed, giving rise to some analyte losses [38,39]. Then, 1-heptanol and 1-undecanol, with a density lower than that of water used as extraction solvents, not only provided higher absorbance than carbon tetrachloride, but was also easier to operate. In UADLLME, after centrifugation of the cloudy solution, 1-heptanol and 1-undecanol was accumulated at the top of the aqueous phase. Then, a pipette was inserted into the floated organic solvent layer, which was easily collected and used for analysis. In addition, alcohols as the extraction solvents (e.g., heptanol, undecanol) may be less toxic than chlorinated solvents (e.g., chloroform, carbon tetrachloride). 1-heptanol had higher absorbance and lower viscosity than 1-undecanol, and therefore 1-heptanol was selected as the extraction solvent in subsequent experiments.

The effect of the volume of 1-heptanol in the range 0.3–1.1 mL on the Cd(II) absorption was also investigated and the results are given in Figure 3b. It was found that maximum absorption is obtained when a volume of 0.5 mL is used. Volumes lesser or greater than 0.5 mL gave much lower absorbance. It seems the reduction of absorbance when used an amount of less than 0.5 mL solvent is because the extraction efficiency of Cd(II)-DDTC is low. Furthermore, the floated organic solvent layer is thin, and it is difficult to collect 1-heptanol without sucking out the underlying water by inserting the micropipette into the floated layer. On the other hand, when amounts greater than 0.5 mL are employed, there is also a decrease in absorbance. Increasing the volume of 1-heptanol will increase the accumulated organic solvent at the top of the aqueous phase and improve the extraction efficiency of Cd(II), but the concentration of analyte in the extraction solvent will decrease. In subsequent experiments, the volume of 0.5 mL 1-heptanol was used because it allowed to analyze a smaller quantity of 1-heptanol, without affecting the extraction efficiency.

#### 2.2.2. Effect of Disperser Solvent

The type of disperser solvents is associated with the extraction efficiency of Cd(II). It is crucial that the water-miscible disperser solvent can mix with both the water-immiscible organic extraction solvent and the aqueous phase to guarantee the formation of the emulsion. Disperser solvent plays a critical part in creating the fine droplets, which increases the extensive contact area, thus speeding up the formation of the Cd(II)-DDTC complex and mass transfer between the two phases. The extraction efficiency can be achieved with the increase of the surface area of the droplets. The used disperser solvent must have the capability to disperse the organic extraction solvent into the aqueous phase. In this study, methanol, ethanol, acetonitrile, and acetone were selected as the disperser solvent, and the results in Figure 3c showed that the absorbance of Cd(II) is at a maximum when ethanol was used. Thus, ethanol was selected as the disperser solvent in the experiment.

The variation of the disperser solvent volume from 0.05 to 0.7 mL was studied and the results were presented in Figure 3d. Figure 3d shows that the absorbance signal increased with the increase of the volume of ethanol between 0.05 and 0.1 mL and then decreased with the further increase of volume. Consequently, 0.1 mL of ethanol was chosen for further study. Ultrasonic technique combining DLLME with ultrasound reduces the use of disperser solvent, while the conventional DLLME technique has a high disperser solvent consumption (around 20% disperser in aqueous sample) [36]. Ultrasound is an electromagnetic energy source which contributes to disperse the organic extraction solvent containing DDTC into an aqueous phase to form a stable cloudy solution. As a result, UADLLME technology has the potential to enhance diffusion and solvation, to assist two immiscible liquids to form the emulsion, and to reduce the use of toxic organic solvents that have adverse environmental effects.

#### 2.2.3. Effect of pH

In UADLLME, the pH values of the sample solution affects the present state of cadmium and DDTC in the solution and the formation of the Cd(II)-DDTC complex. Hydrolysis of metal ions was a probable occurrence under the strong alkali condition. Most metal ions exist in the form of cation in neutral or weak alkaline solution, but hydrolyze and form precipitation when the pH is too high [21,40]. Meanwhile, DDTC was unstable in acidic solutions and could decompose to diethylamine and carbon disulfide. For these reasons, the influence of pH was evaluated by varying the values in the range from 5 to 9. As shown in Figure 3e, the absorbance increased when the pH increased from 5 to 6. The further increasing pH from 6 to 8 obtained constant absorbance. The absorbance decreased when the pH was higher than 8. DDTC is a double-dentate ligand with S as the donor atom. As the pH value increases, the negative charge on S atom in DDTC molecule increases, which is conductive to the complex formation of DDTC with Cd(II) ions. However, when the pH value was higher than 8, the cadmium ions hydrolyzed, resulting in a significant decrease in extraction efficiency. Thereby, pH 7.0 was chosen to be the best condition for the subsequent experiments this study.

#### 2.2.4. Effect of Ultrasound Time and Temperature

Ultrasound could accelerate the mass transfer of the analyte Cd(II)-DDTC from aqueous solution into the organic extraction solvent. It was shown that the extraction time could be shortened via application of ultrasound energy in the extraction process [10,19,22]. For this study, the positive effect of ultrasound on the efficiency has also been verified. The time of ultrasound extraction was investigated from 1 to 9 min. The obtained results, as shown in Figure 3f, indicated that the absorbance signal of Cd(II) gradually increased with the increase of ultrasound time from 1 to 5 min and then decreased with a longer ultrasound time. As a result, we used 5 min as the optimized ultrasound extraction time.

The temperature of the mixture in the ultrasound bath was also studied. The impact of ultrasound bath temperature was examined in the range of 20 °C to 70 °C and the results were shown in Figure 3g. The absorbance signal of cadmium increased with increasing bath temperature to 40 °C, as the higher temperature can accelerate the mass transfer and diffusion of analyte. However, when the ultrasound bath temperature was more than 40 °C, the absorbance signal decreased, most likely because the complex reaction of cadmium with DDTC is exothermic and the high temperature hindered the reaction. Meanwhile, the higher temperature would increase the volatility of the organic solvent. Thus, 40 °C was applied for all experiments.

### 2.3. Effect of Potentially Co-Existing Ions

The effect of 13 common co-existing ions (including K^+^, Na^+^, Ca^2+^, Mg^2+^, Cu^2+^, Co^2+^, Fe^3+^, Cr^2+^, Mn^2+^, Zn^2+^, Cl^−^, NO_3_^−^, and SO_4_^2−^) on the extraction and determination of Cd^2+^ by the proposed method was evaluated under the optimized conditions. The interferences are caused by the competition of the co-existing metal ions complexing with DDTC and co-extracting with Cd(II). Sample solutions containing 2 μg/L of Cd(II) with different amounts of co-existing ions were used in the UADLLME procedure, and the results are listed in Table 1. The recoveries of Cd(II) in the presence of these co-existing ions are maintained in the range of 95–105%. Since cadmium is a soft acid and the S-donor ligand is soft base, this result is in perfect accordance with the HSAB principle. According to this theory, common metal ions contained in food samples, such as K^+^, Na^+^, Ca^2+^, and Mg^2+^, which are classified as hard acids, interact only weakly with a soft base, i.e., the S-donor ligands, and consequently would not interfere with the determination of cadmium. Thus, the interference of the co-existing ions was considered to be negligible on the recoveries of Cd(II) under the optimum conditions, indicating that the developed method was capable of analyzing the target element in rice samples.

### 2.4. Analytical Figures of Merit

After optimization of the variables that affect the proposed method, the linearity and correlation coefficients for cadmium were measured as 0.1–55 μg/L and 0.9964, correspondingly. The detection limit (the lowest concentration of analytes that can be detected but not necessarily quantitated, LOD) and the limit of quantification (the lowest concentration of analyte that can be measured, LOQ) for the proposed procedure were calculated using the equations LOD = 3 s/m and LOQ = 10 s/m, separately (s corresponds to the 10 blank solutions standard deviation and m is the slope of calibration curve after extraction). Under the optimized conditions, LOD and LOQ were measured as 0.69 μg/L and 2.08 μg/L. The enrichment factor (EF) was calculated by the ratio between the concentration of analyte in the sedimented phase and the initial concentration of analyte in the aqueous sample. The EF is 31 for a sample volume of 8.0 mL. The reproducibility and repeatability of the developed method were measured with relative standard deviation (RSD), which is determined by analyzing the standard solution containing 2 μg/L of Cd(II)(n = 10), which was 2.30%.

### 2.5. Application of the Proposed Method

The proposed method was applied for the analysis of Cd(II) in three different rice samples (polished rice, brown rice, and glutinous rice) that were obtained from local markets. In order to verify the accuracy of the method, the recovery by the proposed UADLLME method was carried out using the standard addition method. The rice sample solutions were all split into two samples and one sample was added known amounts of 1 μg/mL Cd(II) standard solution before treatment. The concentrations of Cd(II) were determined in both the spiked (C_1_) and unspiked sample (C_2_). The recovery (R%) was calculated using Equation (2), where C_3_ is the concentration of standard added into the spiked sample.
RR% = [(C_1_ − C_2_)/C_3_] × 100.(2)

As shown in Table 2, the results obtained for analysis of these three different rice samples varied from 0.069 to 0.082 µg/g. These results demonstrated that the cadmium concentration was very low in the three rice samples [10]. Results shown in Table 2 indicate that satisfactory recoveries were obtained in the range of 96.7–113.6% for all the studied rice samples. These results indicate that the method has potential for the determination of Cd(II) in food and water samples [2]. The results showed no significant difference with the certified values, which verified the accuracy and reliability of the method.

### 2.6. Comparison of the Proposed Method with Other Methods

The comparison of the proposed method with other reported preconcentration methods for the determination of cadmium followed by FAAS is summarized in Table 3. The results show that the linear range, LOD, and RSD of this proposed method are comparable with other extraction methods. As can be seen in Table 3, the limit of detection of UADLLME-FAAS by using only 8 mL of aqueous sample is better or similar to conventional DLLME and vortex-assisted liquid–liquid microextraction, except for SFODME-SUPRAS, VA-RTIL-DLLME, and SM-DLLME, which processes lower LODs than obtained in this work. Compared with the conventional DLLME, the proposed method has several advantages, such as less usage of the organic solvent, lower RSD values, and short extraction time. All these results indicate that the UADLLME-FAAS is a fast, sensitive, and simple method for the determination of Cd(II) in rice samples.

## 3. Conclusions

In this study, 10 commercially available ligands were performed for comparative study for the determination of cadmium in rice samples by ultrasound-assisted dispersive liquid-liquid microextraction combined with FAAS. Ligand plays an important role in the UADLLME-FAAS method. It affects not only the extraction efficiency of Cd(II) in the UADLLME process but also the atomization degree of the metal complex in the FAAS procedure. In the process of UADLLME, cadmium ions coordinated with ligands through the coordination groups to form hydrophobic metal chelate. The ligands used for extraction of Cd(II) required high efficiency to transfer the analyte to the extraction solvent. In the process of FAAS, the absorbance signal was determined by the number of free cadmium atoms produced by the thermal decomposition of the cadmium-chelate present in the gaseous state after volatilization. DDTC was used as the ligand in this work for high extraction efficiency and good absorbance signal. Because of the strong ability to complex with DDTC and high thermal decomposition of the Cd(II)-DDTC complex in the acetylene-air flame, cadmium had a very high transport efficiency from the aqueous to organic phase and very good absorbance signal in FAAS, which can ensure the sensitivity of analysis.

In conclusion, ultrasound-assisted dispersive liquid-liquid microextraction coupled with flame atomic absorption spectrometry containing appropriate ligand DDTC to form complex with Cd(II) and followed with extraction by 1-heptanol was studied and optimized. The developed method had good tolerance to coexisting metal ions and cadmium could be accurately determined by the standard addition method. This combination gives good accuracy and performs successfully in the determination of cadmium in rice samples. The UADLLME-FAAS technique with appropriate ligand has several advantages, namely low consumption of organic solvent during extraction, less toxic extraction solvent instead of high toxicity chlorinated solvent, high sensitivity, simple and fast operation, good extraction performance leading to short extraction time, low cost, and good repeatability. The evaluation of the target compounds yielded satisfactory recovery, linearity precision, and detection limit. Analytical characteristics and applications in rice samples indicate that this method has good application potential in the analysis of trace cadmium in water and food samples.

## 4. Materials and Methods

### 4.1. Instrumentation

Determination of Cd(II) was carried out by a flame atomic absorption spectrometry (Puxi TAS-990, Beijing, China), equipped with air-acetylene burner. A cadmium hollow cathode lamp was used as the radiation source. The most sensitive wavelength and lamp current implemented for the determination were 228.8 nm and 2 mA. Slit width and air/acetylene flow rate were 0.4 nm and 1700 mL/min. Burner height and nebulizer flow rate were set to optimum values, at which the maximum absorbance was obtained for the standard solution. A digital pH meter (Leici PHS-3C, Shanghai, China) equipped with a glass-combination electrode was used for the pH adjustment. In order to assist the emulsification, an ultrasonic cleaner set with temperature control (WiseClean^®^WUC-A06H, DAIHAN Sdentific Co., Wonju-si, Korea) was used to assist the emulsification. A centrifuge (Anting KA1000C, Shanghai, China) was used to accelerate the phase separation process. The rice samples were dried with an electric thermostatic air-drying oven (Keweiyongxing GZX-550ASB, Beijing, China) and crushed by a high-speed multifunctional mill comminutor (Shenlian SL-300, Zhejiang, China). The concentrations of Cd(II) in the aqueous phase after extraction were determined by an ICP-MS (NexION 2000, Perkin Elmer, Waltham, MA, USA).

### 4.2. Chemicals

All reagents and solvents were used as received without additional purification. The standard stock solution of cadmium (1000 mg/L) was obtained from (National Research Center for Certified Reference Materials, Beijing, China). Working standard solutions were prepared daily by sequential dilution of the standard stock solution with deionized water. The disperser solvents used include methanol (≥99.5%, Fuyu Fine Chemical Co., Ltd., Tianjin, China); ethanol (≥99.7%, Fuyu Fine Chemical Co., Ltd., Tianjin, China); acetonitrile (95%, Fuyu Fine Chemical Co., Ltd., Tianjin, China), and acetone (≥99.5%, Kant Chemical Co., Ltd., Laiyang, China). The extraction agents used include carbon tetrachloride (≥99.5%, Damao Chemical Testing Factory, Tianjin, China), n-heptanol (≥99.5%, Sinopharm Chemical Reagent Co., Ltd., Shanghai, China), and undecanol (≥99.5%, Sinopharm Chemical Reagent Co., Ltd., Shanghai, China).

The ligands used include ammonium pyrrolidine dithiocarbamate (APDC) (99%, Macklin Chemical Reagent Co., Ltd., Shanghai, China); N,N′-bis(salicylidene)ethylenediamine (BSE) (>99%, Sigma-Aldrich Chemicals Ltd., Shanghai, China); sodium diethyldithiocarbamate (DDTC) (99%, Sinopharm Chemical Reagent Co., Ltd., Shanghai, China); O,O-diethyl phosphordithiooic acid ammonium(DDTP) (95%, Alfa Aesa Chemicals Ltd., Shanghai, China); sodium dimethyldithiocarbamate (DMDTC) (98%, Aladdin Chemicals Ltd., Shanghai, China); diethylenetriaminepentaacetic acid (DTPA) (≥99%, Macklin Chemical Reagent Co., Ltd., Shanghai, China); ethylenediamine tetraacetic acid (EDTA) (≥99.5%, Sinopharm Chemical Reagent Co., Ltd., Shanghai, China); nitrilotriacetic acid (NTA) (≥98.5%, Sinopharm Chemical Reagent Co., Ltd., Shanghai, China); and 1-(2-Pyridylazo)-2-naphthol (PAN) (98%, Macklin Chemical Reagent Co., Ltd., Shanghai, China). The solutions of 0.001 M PAN, BSE, and NTA were prepared by dissolving the appropriate amount of corresponding compounds in methanol separately. The remaining 0.001 M ligand solutions were prepared by dissolving the appropriate amount of ligands in deionized water, respectively.

In order to decrease the possibility of metal contamination, all glassware and plastic bottles used were cleaned by rinsing with deionized water, soaking with nitric acid solution (10%, *v/v*) for 24 h and then rinsing three times with deionized water before use.

### 4.3. Optimization of UADLLME-FAAS Procedure

The 8 mL of standard solution containing 10 μg/L Cd(II) or the rice sample solution was transferred into a 10 mL glass conical bottom tube with screw cape. The pH was adjusted to 7.0 with dilute hydrochloric acid or ammonia water. Subsequently, 0.8 mL 0.001 M DDTC solution as the complexing agent, 0.5 mL of 1-heptanol as the extraction solvent, and 0.1 mL of ethanol as the disperser solvent were added into the tube. The mixture was ultrasonicated for 5 min at 40 °C. Then, the solution was centrifuged at 3000 rpm for 5 min to get better separation of the two phases. A total of 0.3 mL organic phase containing the Cd(II)-DDTC complex was taken by micropipette and mixed with 0.4 mL of methanol. Then, FAAS was used for the determination of Cd(II).

### 4.4. Co-Existing Ions Studies

Potentially co-existing ion solutions were prepared from various concentrations of cations and anions traditionally present in food samples. These included K^+^, Na^+^, Ca^2+^, Mg^2+^, Cu^2+^, Co^2+^, Fe^3+^, Cr^2+^, Mn^2+^, Zn^2+^, Cl^−^, NO_3_^−^, and SO_4_^2−^ at concentrations of 1, 2, 3, and 4 mg/L added to 2 μg/L of Cd^2+^. The signals obtained were compared to those with no interfering ions.

### 4.5. Preparation of Rice Samples

Rice samples, including polished rice, brown rice, and glutinous rice, were purchased from local markets. Each rice sample was thoroughly mixed and placed in an oven at 100 °C for 2 h. Then, 10 g of each rice sample was ground using high-speed multifunctional mill at a speed of 25,000 r/min for 1 min, sifted with 100-mesh stainless steel sifter, and kept for the digestion step. For sample digestion, 0.1 g powdered sample was accurately weighed into a conical flask with a small funnel as cover. A total of 10 mL of nitric acid-percholoric acid mixture (V:V, 9:1) was added and soaked for 24 h. Then, the sample was slowly heated on a hot plate until the nitric acid-percholoric acid mixture was volatilized completely. The residue was dissolved in 0.1 M nitric acid, transferred into a 100 mL volumetric flask, and diluted to volume with deionized water. The clear solution, without any visible residue, was adjusted to a pH of 7 with dilute ammonia water and transferred into pre-cleaned polyethylene screw capped bottles and stored in a refrigerator at 4 °C prior to the UADLLME-FAAS analytical process.

## Figures and Tables

**Figure 1 molecules-27-00590-f001:**
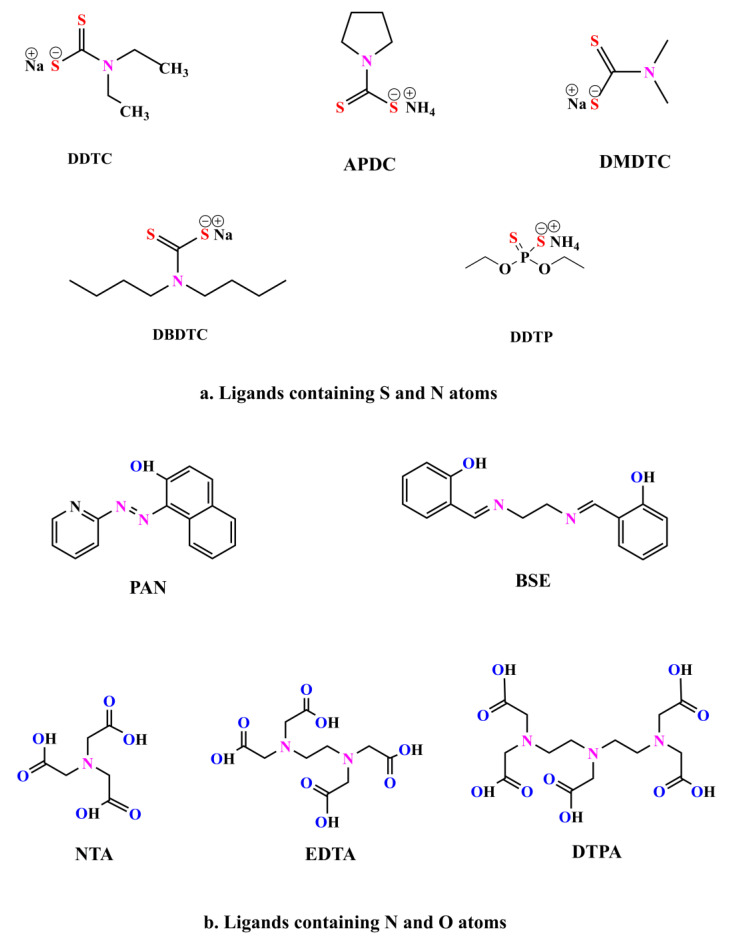
The chemical structural formulas of the ligands.

**Figure 2 molecules-27-00590-f002:**
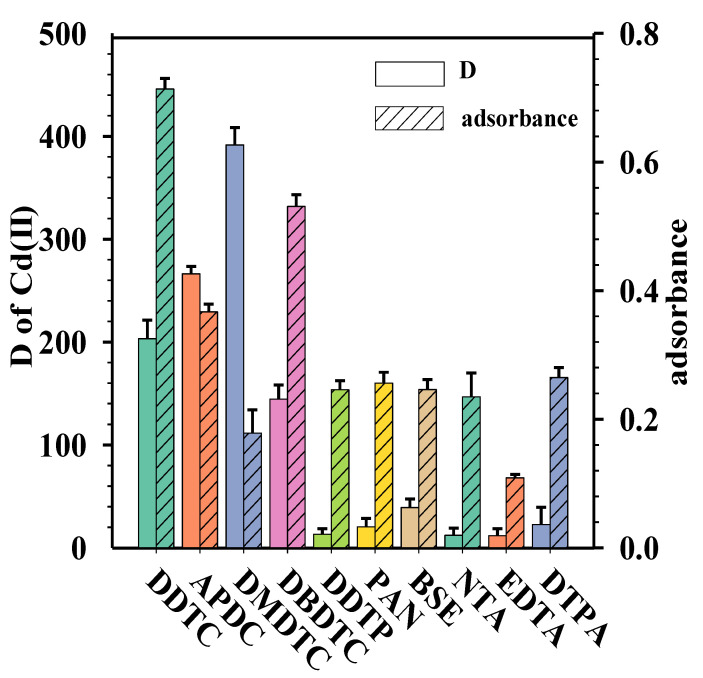
The effect of the types of ligands (the error bars are the standard deviation for n = 3). Conditions: volume of standard solution containing 10 μg/L Cd(II), 8 mL; volume of solution containing 0.001 M ligands, 0.8 mL; volume of 1-heptanol, 0.5 mL; volume of ethanol, 0.1 mL; sample pH, 7.0; ultrasound extraction time, 5 min; ultrasound temperature 40 °C; centrifuging for 5 min at 3000 rpm.

**Figure 3 molecules-27-00590-f003:**
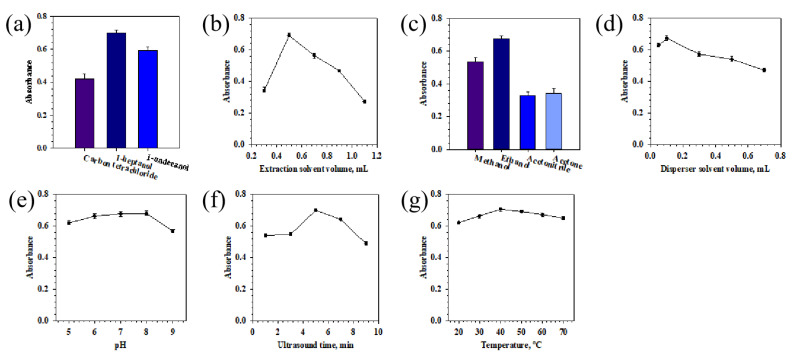
Optimized parameters of the UADLLME procedure (standard deviation for n = 3). (**a**) Types of extraction solvent; (**b**) volume of extraction solvent (mL); (**c**) types of dispersive solvent; (**d**) volume of dispersive solvent (mL); (**e**) pH; (**f**) ultrasound time; (**g**) ultrasound temperature. Conditions: volume of standard solution containing 10 μg/L Cd(II), 8 mL; volume of solution containing 0.001 M DDTC, 0.8 mL; centrifuging for 5 min at 3000 rpm; (**a**) volume of ethanol 0.1 mL, sample pH 7.0, ultrasound extraction time 5 min, ultrasound temperature 40 °C, 0.5 mL of extraction solvent (carbon tetrachloride, 1-heptanol, and 1-undecanol); (**b**) volume of ethanol 0.1 mL, sample pH 7.0, ultrasound extraction time 5 min, ultrasound temperature 40 °C, 0.3–1.1 mL of extraction solvent (1-heptanol); (**c**) volume of 1-heptanol 0.5 mL, sample pH 7.0, ultrasound extraction time 5 min, ultrasound temperature 40 °C, 0.1 mL of disperser solvent (methanol, ethanol, acetonitrile, and acetone); (**d**) volume of 1-heptanol 0.5 mL, sample pH 7.0, ultrasound extraction time 5 min, ultrasound temperature 40 °C, 0.05–0.7 mL of disperser solvent (ethanol); (**e**) volume of 1-heptanol 0.5 mL, volume of ethanol 0.1 mL, ultrasound extraction time 5 min, ultrasound temperature 40 °C, sample pH range 5–9; (**f**) volume of 1-heptanol 0.5 mL, volume of ethanol 0.1 mL, sample pH 7.0, ultrasound temperature 40 °C, sample pH: 7.0, ultrasound time 1–9 min; (**g**) volume of 1-heptanol 0.5 mL, volume of ethanol 0.1 mL, sample pH 7.0, ultrasound extraction time 5 min, ultrasound temperature 20–70 °C.

**Table 1 molecules-27-00590-t001:** Effect of co-existing ions on the determination of cadmium.

Interference	Added as	Ratio of Co-Existing Ions to Cd(II) (*w/w*)	Recovery (%), Mean ± SD ^a^ (n = 3)
K^+^	KCl	2000	96.3 ± 5.1
Na^+^	NaCl	2000	96.1 ± 4.6
Ca^2+^	CaCl_2_	1500	95.8 ± 5.1
Mg^2+^	MgCl_2_	2000	92.7 ± 5.6
Cu^2+^	CuSO_4_	500	93.4 ± 8.4
Co^2+^	Co(NO_3_)_2_	1500	92.9 ± 6.5
Fe^3+^	FeCl_3_	1500	96.4 ± 2.1
Cr^3+^	Cr(NO_3_)_3_	500	95.0 ± 4.8
Mn^2+^	Mn(NO_3_)_2_	1000	95.4 ± 5.2
Zn^2+^	Zn(NO_3_)_2_	1500	94.3 ± 4.4
Cl^−^	NH_4_Cl	2000	95.6 ± 4.3
NO_3_^−^	NaNO_3_	2000	95.7 ± 4.2
SO_4_^2^^−^	Na_2_SO_4_	2000	94.9 ± 4.7

^a^ standard deviation.

**Table 2 molecules-27-00590-t002:** Analytical results for determination of Cd(II) in the rice samples.

Samples	Added Standards of Cd(II) (μg/L)	Found, Mean ± SD (n = 3) (μg/g)	Relative Recovery (%)
polished rice	0	0.069 ± 0.013	-
	1	1.127 ± 0.021	105.8
	2	2.046 ± 0.045	98.8
brown rice	0	0.070 ± 0.018	-
	1	1.097 ± 0.022	102.7
	2	2.343 ± 0.064	113.6
glutinous rice	0	0.082 ± 0.025	-
	1	1.154 ± 0.036	107.2
	2	2.016 ± 0.040	96.7

**Table 3 molecules-27-00590-t003:** Comparison of the proposed method with other extraction methods for the determination of Cd(II) followed by FAAS.

Extraction Method	Linear Range (μg/L)	LOD (μg/L)	RSD (%)	Ref.
^a^ In situ-TSIL-DLLME	5–250	0.55	1.2	[2]
^b^ FFSE	1.2–60	0.30	2.8	[11]
^c^ SFODME-SUPRAS	5–700	0.09	2.7–3.9	[13]
DLLME	1.0–50	0.74	3.2	[15]
DLLME	5.0–100	1.3	7.6	[16]
^d^ VA-RTIL-DLLME	1.0–225	0.25	2.4	[18]
^e^ UA-CPE	3–250	0.9	3.6	[41]
^f^ SM-DLLME	5–180	0.3	4.2	[42]
^g^ SPE	-	0.81	<10	[43]
^h^ VALLME	10–250	2.9	4.1	[44]
^i^ HFRLM	5–30	1.5	4	[45]
^j^ CPE	2.0–100	0.70	3.2	[46]
^j^ CPE	1–100	0.44	0.99	[47]
UADLLME	0.1–55	0.69	2.30	This work

^a^ In situ task-specific ionic liquid dispersive liquid-liquid microextraction; ^b^ fabric fiber sorbent extraction; ^c^ solidification floating organic drop microextraction based on supramolecular solvent; ^d^ vortex-assisted room temperature ionic liquid dispersive liquid-liquid microextraction; ^e^ ultrasonic-assisted cloud point extraction; ^f^ supramolecular-based dispersive liquid-liquid microextraction; ^g^ solid phase extraction; ^h^ vortex-assisted liquid-liquid microextraction; ^i^ hollow fiber renewal liquid membrane; ^j^ cloud point extraction.

## Data Availability

Data is contained within the article.

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
