# Peer review of "Comparison Studies on Several Ligands Used in Determination of Cd(II) in Rice by Flame Atomic Absorption Spectrometry after Ultrasound-Assisted Dispersive Liquid–Liquid Microextraction"

_molecules, 2022, doi:10.3390/molecules27030590_

Round 1

Reviewer 1 Report

The comment and suggestions against the work are given below. Hope all concerns help the author improve the work before reconsidering publication. 1. L10,12: ligands are normally used as complexing agents for metal derivatization or to form complex, but not used for preconcentration. Metal-ligand complex maybe affects the efficiency of extraction due to its property change, e.g., hydrophobicity/hydrophilicity. So, the authors should revise carefully. 2.L17: disperser solvent / dispersive solvent is normally used. 3. L22: "between 95.6 and 100.8%" However, these %R values were wrong." Needs correction. 4.L23: was successfully applied ..... 5. L40: Please revise "control the content". It seems to be not suitable for this content, try to find other suitable words. 6. The authors would like to present various ligands to form complexation, but the characteristic of each ligand is missing. As we know, the advantages/disadvantages/specific condition for each ligand is the main point to concerns. Please revise this content on the Introduction part. 7. Why did the authors select UA-DLLME as the extraction/preconcentration method? Sometimes, typical DLLME may be used and it is simpler than the proposed work. This point is needed in the Introduction Part. 8. What is the different point/novelty of the present work? Sometimes, the comparison to other related works should be provided properly. 9. L92: Please correct the term as shown in Eq.(1) 10. Figure 2: The labeled "Adsorbance" should be changed to absorbance. 11. According to Figure 2, the D value and Absorbance are need described in more detail. For example, DDTC provided D values less than APDC, but absorbance was higher. This observation is also found for most ligands. So, the authors should provide/describe more detail completely. Please provide some comments on the relation of D value and absorbance results. 12. L114-116: It seems to be incorrect. Please revise properly based on the results in Figure 2. 13. Why did the authors select DDTC comparison to APDC? Although the authors gave some reasons as in Line 118 already, more information is still needed to describe properly. 14. L141-144: We know that the use of low-density solvent as extraction solvent seems to be difficult to separate the extraction phase, normally it needs a special device. So, the authors should revise properly or find some other advantage to support this statement. 15. L146: Figure 3a 16. LL167: Figure 3b 17. L186: Figure 3c 18. L200: Figure 3d 19. L205: Figure 3e 20. The authors should provide the detail involving the results obtained such as the trend of the graph increase or decrease, try to give reasonable information. 21. L252-253: The %R is not corrected. Please re-calculate based on the result shown in Table 2 22. Please correct the %R in Table 2. All results are wrong. Please also correct the significant figures. For example, -polished rice: [(1.118-0.085)/1.000]*100=103.3%; [(2.000-0.085)/2.000]*100=95.8% -brown rice: 92.8 and 121.4% -glutinous rice: 100.5 and 94.7% 23. Please comment on the results obtained in Table 2, Are they acceptable values? Sometimes, the references may be cited to support this point. 24. Results and discussion should be merged. It seems to be easy to understand and more formal. 25. Conclusions should be separated. As the authors provided, it should be revised the conclusion. 26. Would like to know about the information of ICP-MS used in this study? The results obtained from ICP-MS can be used to compare with the proposed method?

Author Response

Dear Editors and Reviewers:

On behalf of my co-authors, we thank you very much for giving us an opportunity to revise our manuscript, we appreciate editors and reviewers very much for their positive and constructive comments and suggestions on our manuscript entitled “Comparison studies on several ligands used in determination of Cd (II) in rice by flame atomic absorption spectrometer after ultrasound-assisted dispersive liquid–liquid microextraction” (ID: molecules-1515078). Those comments are all valuable and very helpful for revising and improving our paper, as well as the important guiding significance to our researches. We have studied comments carefully and have made correction which we hope meet with approval. Revised portion are marked up using the “Track Changes” function in the paper. The main corrections in the paper and the point-to-point responds to the reviewer’s comments are as flowing: 

Reply to Reviewer #1

Point 1:

L10,12: ligands are normally used as complexing agents for metal derivatization or to form complex, but not used for preconcentration. Metal-ligand complex maybe affects the efficiency of extraction due to its property change, e.g., hydrophobicity/hydrophilicity. So, the authors should revise carefully.

Response 1:

We feel great thanks for your professional review work on our article. According to your suggestions, we have made corrections to our previous draft, all of the inappropriate expressions were corrected.   

Point 2:

L18: disperser solvent / dispersive solvent is normally used.

Response 2:

Thanks for your very thoughtful suggestion. We have changed "dispersing agent" to "disperser solvent".

Point 3:

L23: "between 95.6 and 100.8%" However, these %R values were wrong." Needs correction.

Response 3:

We are really sorry for our mistakes. The %R values were corrected according to your comments.

Point 4:

L24: was successfully applied .....

Response 4:

We feel sorry for our writing mistake. In our resubmitted manuscript, the mistake was revised. Thanks for your correction.

Point 5:

L44: Please revise "control the content". It seems to be not suitable for this content, try to find other suitable words.

Response 5:

Thanks for your suggestion. We have changed "control the content" to "monitor the concentration".  .

Point 6:

The authors would like to present various ligands to form complexation, but the characteristic of each ligand is missing. As we know, the advantages/disadvantages/specific condition for each ligand is the main point to concerns. Please revise this content on the Introduction part.

Response 6:

Thank you for your comment. We have added the suggested content to the manuscript and shown as below.

Ligands with completely different chemical structures and properties have been used in studies of the determination of cadmium as mentioned above. Dithiocarbamate (DTC) and dithiophosphate(DTP) are two well known groups of compounds that may bind strongly and selectively with cadmium ions and forming Cd(II)-ligand compounds. Both of DTC and DTP can form chelates with Cd(II) through the two donor sulfur atoms. The tow groups of compounds can coordinate symmetrically with Cd(II) by involving the both sulfur atoms, as well as asymmetrically by involving only one sulfur atom. However, the extraction conditions of Cd(II) by using DTC and DTP are quite different. For example, acidic conditions are suggested to enhance the extraction efficiency of DDTP, while DDTC decompose into diethylamine and carbon disulfide in acidic solution[35]. Even APDC, DMDTC and DBDTC have similar chemical structures, the stability of the complexes with Cd(II) which affect the FAAS procedure are different due to their different substituents.

Point 7:

Why did the authors select UA-DLLME as the extraction/preconcentration method? Sometimes, typical DLLME may be used and it is simpler than the proposed work. This point is needed in the Introduction Part.

Response 7:

Thank you for your comment. We have added the suggested content to the manuscript and shown as below.

Compared with the typical DLLME procedures, employing a ultrasound approach to this method led to some advantages that can be summarized as follows: decrease of disperser solvent usage, the alternating acoustic pressure speeds up the extraction procedure, and increase of extraction efficiency [36].

Point 8:

What is the different point/novelty of the present work? Sometimes, the comparison to other related works should be provided properly.

Response 8:

Thank you for your valuable suggestion. We have added the suggested content to the manuscript and shown as below as well.

Although in the mentioned DLLMEs, the presence of an appropriate complexing agent for extraction of Cd(II) is necessary[2, 12, 15-19, 22-27], there are rare discussions about the effect of types of ligand which plays an important role in DLLMEs. Most of previous researchers demonstrated that the types of extraction solvent and disperser solvent were important in the DLLMEs technique. Actually, the organic ligands play an important role in determination of trace Cd(II), because they are affect not only the extraction efficiency of Cd(II) but also the FAAS behaviour of metals. In this paper, a comparative study for the extraction efficiency of Cd(II) and atomization degree of Cd(II)-ligand complex by using ten commercially available ligands was performed. 

Point 9:

L114,115: Please correct the term as shown in Eq.(1) 

Response 9:

Thank you for your comment. We have carefully checked the manuscript and corrected the term as shown in Eq.(1).

Point 10:

Figure 2: The labeled "Adsorbance" should be changed to absorbance.

Response 10:

We have corrected it. Thank you for your pointing out.

Point 11:

According to Figure 2, the D value and Absorbance are need described in more detail. For example, DDTC provided D values less than APDC, but absorbance was higher. This observation is also found for most ligands. So, the authors should provide/describe more detail completely. Please provide some Points on the relation of D value and absorbance results.

Response 11:

Thank you for your comments. We have added the suggesting contents in the revised manuscript and listed as below as well.

From Figure 2, it can be seen that the ligands containing sulfur atoms (DDTC, APDC, DMDTC, DBDTC and DDTP) had higher extraction efficiency than the five N-donor ligands (PAN, BSE, NTA, EDTA and DTPA) under the current experimental conditions. For the five N-donor ligands, the strength and content of hydrophilic groups have significant impacts on the extraction behavior of cadmium. The distribution coefficient was decreased by increasing the hydrophilic group strength and content. For the five S-donor ligands, the higher D values of S-donor ligands were directly related to two donor sulfur atoms. According to the hard-soft acids-bases (HSAB) principle, Cd (II) is a reasonably soft metal ion, which prefers to coordinate with ligands containing N-donor atoms or S-donor groups. Meanwhile, according to HSAB, sulfur atom which is classified as a softer base than nitrogen atom and thus have stronger interaction with the soft acid Cd (II) ions. Therefore the ligands containing sulfur atoms have higher D of Cd(II).

However, highly efficiency extraction is important but not necessary in the FAAS process.For example, DMDTC provided the highest D in the ten studied ligands but absorbance signal was low. In contrast with DMDTC, although DDTC provides D values less than APDC and DMDTC, the absorbance was the highest in the ten ligands. The reason is that organic complexing reagents exhibit various effects in FAAS.

Obviously, the four alkyldithiocarbamates (R2N)C(S)S- have much higher D than alkylthiophosphate (RO)2P(S)S- (DDTP), which can be explained on the basis of the electronic property of the substituent groups.

The four alkyl-dithiocarbamate compounds, ie, DDTC, APDC, DMDTC and DBDTC, only Cd(II)-DDTC exhibited high volatility and high degree of atomization of cadmium in acetylene flame and produced very good absorbance signal of cadmium. This maybe because DMDTC and APDC formed stronger and more thermally stable cadmium-chelate than DDTC.

Point 12:

L114-116: It seems to be incorrect. Please revise properly based on the results in Figure 2.

Response 12:

We are sorry for our negligence. L113 to L118 was corrected in the revised manuscript. Thank you for pointing this out.

Point 13:

Why did the authors select DDTC comparison to APDC? Although the authors gave some reasons as in Line 118 already, more information is still needed to describe properly.

Response 13:

Thank you for your comments. DDTC and APDC have been widely used as complexing agents in extraction and determination of metal ions. Based on our results, although DDTC and APDC have the similar chemical structure and both have high extraction efficiency for Cd(II), the volatility and degree of atomization of Cd(II)-DDTC and Cd(II)-APDC in acetylene flame were very different. Obviously, Cd(II)-DDTC exhibited high volatility and high degree of atomization, therefore produced very good absorbance signal of cadmium.

Point 14:

L141-144: We know that the use of low-density solvent as extraction solvent seems to be difficult to separate the extraction phase, normally it needs a special device. So, the authors should revise properly or find some other advantage to support this statement.

Response 14:

Thank you for your valuable suggestions to improve the quality of our manuscript. We have added the suggested content to the manuscript and shown as below as well.

 Carbon tetrachloride has the lowest absorbance signal seems to because in the presence of chlorinated extraction solvent, volatile chlorides of Cd(II) can be formed, giving rise to some analyte losses [38,39]. 1-heptanol and 1-undecanol with a density lower than that of water used as extraction solvents not only provided higher absorbance than carbon tetrachloride, but also was easier to operate. In UADLLME, after centrifugation of the cloudy solution, 1-heptanol and 1-undecanol was accumulated at the top of the aqueous phase. Then pipette was inserted into the the floated organic solvent layer which was easily collected and used for analysis. 1-heptanol had higher absorbance and lower viscosity than 1-undecanol, 1-heptanol was selected as the extraction solvent in subsequent experiments.

Point 15:

L178: Figure 3a ; L206: Figure 3b ; L225: Figure 3c ; L240: Figure 3d ; L246: Figure 3e: The authors should provide the detail involving the results obtained such as the trend of the graph increase or decrease, try to give reasonable information.

Response 15:

Thanks for your careful checks. We have made the corrections in the revised manuscript. We have tried our best to give reasonable information and hopefully you will find this revised version satisfactory.

Point 16:

L297: The %R is not corrected. Please re-calculate based on the result shown in Table 2.

Response 16:

Thank you for your comments. We feel sorry for our carelessness. We have re-calculate the %R based on Table 2.

Point 17:

Please correct the %R in Table 2. All results are wrong. Please also correct the significant figures. For example, -polished rice: [(1.118-0.085)/1.000]*100=103.3%; [(2.000-0.085)/2.000]*100=95.8% -brown rice: 92.8 and 121.4% -glutinous rice: 100.5 and 94.7%

Response 17:

We are extremly sorry for our negligence. We have re-calculated the %R according to your suggestion.

Point 18:

Please Point on the results obtained in Table 2, Are they acceptable values? Sometimes, the references may be cited to support this point.

Response 18:

We gratefully appreciate for your valuable suggestion. We have added the suggested contents in the revised manuscript and shown as below as well.

Results shown in Table 2 indicate that satisfactory recoveries were obtained in the range of 96.7–113.6% for all the studied rice samples. These results indicate that the method has potential for the determination of Cd(II) in food and water samples[41].

Point 19:

Results and discussion should be merged. It seems to be easy to understand and more formal.

Response 19:

We are very sorry for the inconvenience brought to reviewer in your reading. The manuscript has been thoroughly revised and results and discussion were merged, so we hope it can meet the journal’s standard. Thanks so much for your useful comments.

Point 20:

Conclusions should be separated. As the authors provided, it should be revised the conclusion.

Response 20:

We gratefully appreciate for your valuable suggestion. We have separated and revised the conclusions in the revised manuscript and shown as below as well.

In this study, ten commercially available ligands were performed for comparative study for determination of cadmium in rice samples by ultrasound-assisted dispersive liquid–liquid microextraction combined with FAAS. Ligand plays an important role in UADLLME-FAAS method. It affects not only the extraction efficiency of Cd(II) in UADLLME process but also the atomization degree of metal complex in FAAS procedure. In the process of UADLLME, cadmium ions coordinated with ligands through the coordination groups to form hydrophobic metal chelate. The ligands used for extraction of Cd (II) required high efficiency to transfer the analyte to the extraction solvent. In the process of FAAS, the absorbance signal was determined by the number of free cadmium atoms produced by the thermal decomposition of the cadmium-chelate presents in the gaseous state after volatilization. DDTC was used as ligand in this work for high extraction efficiency and good absorbance signal. Because of the strong ability to complex with DDTC and high thermal decomposition of Cd(II)-DDTC complex in the acetylene-air flame, cadmium had a very high transport efficiency from aqueous to organic phase and very good absorbance signal in FAAS that can ensure the sensitivity of analysis.

In conclusion, ultrasound-assisted dispersive liquid–liquid microextraction coupled with flame atomic absorption spectrometry containing appropriate ligand DDTC to form complex with Cd(II) and followed with extraction by 1-heptanol was studied and optimized. The developed method had good tolerance to coexisting metal ions and cadmium can be accurately determined by the standard addition method. This combination gives good accuracy and performs successfully in the determination of cadmium in rice samples. The UADLLME-FAAS technique with appropriate ligand has several advantages namely low consumption of organic solvent during extraction, high sensitivity, simple and fast operation, good extraction performance lead to short extraction time, low cost and good repeatability. The evaluation of the target compounds yielded satisfactory recovery, linearity precision and detection limit. Analytical characteristics and applications in rice samples indicate this method has a good application potential in the analysis of trace cadmium in water and food samples.

Point 21:

Would like to know about the information of ICP-MS used in this study? The results obtained from ICP-MS can be used to compare with the proposed method?

Response 21:

We are appreciative of the reviewer’s comment. Indeed, it will be more interesting if we get a comparative assessment of the results obtained by ICPMS and FAAS. However, the ICPMS of our college only supports the measurement of metal ion content in water samples. Therefore we have not employed ICPMS for the determination of Cd(II) in the organic extraction solvent used in DLLME.

In our study, we have used ICPMS in the comparptive studies of ten ligands. It was employed to study the microextraction efficiency of the ten ligands for Cd(II). Several ligands shown high extraction efficiencies of Cd(II) and the residual Cd(II) concentration in the water phase after UADLLME was very low. In order to obtain more accurate distribution coefficient results, we used ICPMS to measure the content of Cd(II) in water phase.

Reviewer 2 Report

The manuscript „Comparison studies on several ligands used in determination of Cd (II) in rice by flame atomic absorption spectrometer after ultrasound-assisted dispersive liquid–liquid microextraction“ by Q. Sun and co-workers reports on the development and optimization of a UADLLME method (considering colume of extract solvent and disperser, pH value, ultrasonic time and temperature) for the subsequent analysis of cadmium(II) using FAAS. The elaborated method was applied to quantify cadmium(II) in three different samples of rice. The method showed good analytical parameters to other established methods and also in the presence of other frequently occurring ions. The novel method contributes to the analysis of toxic metals in food.

It is recommended to consider a more recent Review on Cadmium’s Toxicity in the introduction: https://doi.org/10.3390/ijerph17113782

„graphite furnace atomic absorption spectrometry (GFAAS) [19-22], electrothermal atomic absorption spectrometry (ETAAS) [23]“: I am wondering the difference between GFAAS and ETAAS, because the authors of literature [23] also use GFAAS. Actually, the electrothermal treatment of the sample in order to atomize occurs in the GF and therefore it is equal method.

“and inductively coupled plasma mass spectrometry (ICP-MS) [25-28].” ICPMS seem to be the method of choice for the determination of metal among science, but the authors decided to choose AAS anyways. The authors should provide a reference for the advantages and superiority of AAS compared to ICPMS, for example as discussed in: 10.1039/c9dt03330k

“Dispersive liquid–liquid microextraction (DLLME) is an attractive method” The authors are kindly suggested to consider a very good publication on DLLME: https://doi.org/10.1016/j.trac.2011.04.014

The chemical structures of Figure 1 are very small and exhausting to look at them. Please provide better resolution. In Figure 2, the authors should provide, if the mean values are shown and the number of measurements (n=?). How about standard deviation or standard error of the means?

Is eciency = efficiency?

Figure 3(e): Temperature, °C; please provide the number of measurements (n=?) in the caption of the Figure.

“The effect of the volume of 1-heptanol in the range of 0.3-1.1 mL on the Cd (II) absorption was also investigated and the results are given in Fig. 1.” This result is shown in Figure 3a. Can the authors explain why, first, increase of volume is beneficial up to 0.5 ml, and then there is a decrease.

“In this study, methanol, ethanol, acetonitrile and acetone were selected as disperser solvent, and the results showed the absorbance of Cd is at a maximum when ethanol was used.” Where are the results of the other disperser solvents?

“The variation of the disperser solvent volume from 0.05 to 0.7 mL was studied and the results were presented in Fig. 2. Fig. 2” It is Figure 3(b).

“As shown in Fig. 3, the absorbance increased when the pH increased from 5 to 7.” Figure 3(c). Actually it is nearly the equal value of pH 6, pH 7, pH 8.

“The obtained results as 199 shown in Fig. 4”. It is Figure 3(d) “and the results were shown in Fig. 5.” This is Figure 3(e).

Actually, it is not cadmium metal Cd, but cadmium ion (Cd2+) or Cd(II).

Table 1: Is the values from one single measurement? Or did the authors performed more measurements (n=?). It is kindly suggested to do more than one measurement and to provide the mean and the standard deviation or standard error.

Table 2: Found +/-SD: no SD data provided. Please add. How about the number of experiments (n=?)

How about the values of cadmium in the three different rice samples without added standrads (0 microg/L) regarding toxicity thresholds? Please judge on the obtained values.

Table 3: Can the authors also include other works: https://doi.org/10.1016/j.measurement.2019.07.069 from 2019, https://doi.org/10.1007/s13738-011-0018-7, 2017 Journal of Chemical, Biological and Physical Sciences 7(3):578-587? How was the selection of the examples of Table 3 based on by the authors?

In conclusion, The authors should stress the advantage of their study compared to the other studies of table 3. What is the advantage of the UADLLME compared to the other extraction styles?

Author Response

Point 1:

The manuscript „Comparison studies on several ligands used in determination of Cd (II) in rice by flame atomic absorption spectrometer after ultrasound-assisted dispersive liquid–liquid microextraction“ by Q. Sun and co-workers reports on the development and optimization of a UADLLME method (considering colume of extract solvent and disperser, pH value, ultrasonic time and temperature) for the subsequent analysis of cadmium(II) using FAAS. The elaborated method was applied to quantify cadmium(II) in three different samples of rice. The method showed good analytical parameters to other established methods and also in the presence of other frequently occurring ions. The novel method contributes to the analysis of toxic metals in food.It is recommended to consider a more recent Review on Cadmium’s Toxicity in the introduction: https://doi.org/10.3390/ijerph17113782

Response 1:

Thank you for your introduction to the wonderful research work. According to your suggestion, we properly cite this article as reference [1].

Point 2:

graphite furnace atomic absorption spectrometry (GFAAS) [19-22], electrothermal atomic absorption spectrometry (ETAAS) [23]“: I am wondering the difference between GFAAS and ETAAS, because the authors of literature [23] also use GFAAS. Actually, the electrothermal treatment of the sample in order to atomize occurs in the GF and therefore it is equal method.

Response 2:

Thank you for your comment. We agree with the reviewer that GFAAS and ETAAS are equal methods. Literature[23] had been incorporated into classification GFAAS.

Point 3:

“and inductively coupled plasma mass spectrometry (ICP-MS) [25-28].” ICPMS seem to be the method of choice for the determination of metal among science, but the authors decided to choose AAS anyways. The authors should provide a reference for the advantages and superiority of AAS compared to ICPMS, for example as discussed in: 10.1039/c9dt03330k

Response 3:

It is really a great suggestion as the reviewer pointed out. We have carefully read the literature 10.1039/c9dt03330k, unfortunately, we did not find the relevant content. We had rewrite the advantages of FAAS compared to ICPMS and cited two references to provide support for our choice of FAAS in the revised manuscript and shown below as well. We hope these references are supportable for the advantages of FAAS compared to ICPMS.

Compared with costly analysis technique which are rarely found in laboratories of routine analysis , such as ICP-OES and ICPMS, FAAS is a conventional technique which has been widely applied in metals analysis because it is a less expensive and simpler alternative technique[29, 30]. In addition, FAAS is faster, easier to operate and has considerable accuracy and sensitivity. Laboratories are unlikely to make the switch to ICP-MS or ICP-OES since the analysis can be performed using the existing FAAS methods, especially when the laboratory focus on single element analysis, such as that for Cd(II) in rice.

[29]10.2478/v10007-011-008-4

[30] http://dx.doi.org/10.1016/j.marpolbul.2016.10.068

Point 4:

“Dispersive liquid–liquid microextraction (DLLME) is an attractive method” The authors are kindly suggested to consider a very good publication on DLLME: https://doi.org/10.1016/j.trac.2011.04.014

Response 4:

Thank you for your introduction to the wonderful research work. According to your suggestion, we properly cite this article as reference [32]. 

Point 5:

The chemical structures of Figure 1 are very small and exhausting to look at them. Please provide better resolution.

Response 5:

Thank you for your comment. We have provide Figure 1 with a better resolution in the revised manuscript and shown below as well.

Figure 1. The chemical structural formulas of ligands

Point 6

In Figure 2, the authors should provide, if the mean values are shown and the number of measurements (n=?). How about standard deviation or standard error of the means?

Response 6:

Thank you for your comment.We have made corrections according to the reviewer’s comments in the revised manuscript and shown below as well.

Figure 2. The effect of the types of ligands(standard deviation for n=3).

Point 7:

Is eciency = efficiency?

Response 7:

We are extremly sorry for our negligence. We have corrected the mistake. Thank you for pointing this out.

Point 8:

Figure 3(e): Temperature, °C; please provide the number of measurements (n=?) in the caption of the Figure.

Response 8:

Thank you for your comment.We have made corrections according to the reviewer’s comments and provided the number of measurements (n=3) in the caption of Figure 3.

Point 9:

“The effect of the volume of 1-heptanol in the range of 0.3-1.1 mL on the Cd (II) absorption was also investigated and the results are given in Fig. 1.” This result is shown in Figure 3a. Can the authors explain why, first, increase of volume is beneficial up to 0.5 ml, and then there is a decrease.

Response 9:

Thank you for your comments. We have added the suggested content in the revised manuscript and shown as below as well.

It seems the reduction of absorbance when used an amount of less than 0.5 mL solvent is because the extraction efficiency of Cd(II)-DDTC is low. Furthermore, the floated organic solvent layer is thin and it is difficult to collect 1-heptanol without sucking out the underlying water by inserting the micropipette into the floated layer. On the other hand, when amounts greater than 0.5 mL are employed, there is also a decrease in absorbance. Increasing the volume of 1-heptanol will increase the accumulated organic solvent at the top of the aqueous phase and improve the extraction efficiency of Cd (II), but the concentration of analyte in the extraction solvent will decrease.

Point 10:

“In this study, methanol, ethanol, acetonitrile and acetone were selected as disperser solvent, and the results showed the absorbance of Cd is at a maximum when ethanol was used.” Where are the results of the other disperser solvents?

Response 10:

Thank you for your comment. The results of the other disperser solvents have been provided as Figure 3(c) while results of the other extraction solvents provided as Figure 3(a) in the revised manuscript and shown below as well.

Figure 3. Optimized parameters of the UADLLME procedure(standard deviation for n=3): (a) types of extraction solvent (b) volume of extraction solvent (mL) (c) types of dispersive solvent (d) volume of dispersive solvent (mL) (e) pH (g) and (h) ultrasound time and temperature.

Point 11:

“The variation of the disperser solvent volume from 0.05 to 0.7 mL was studied and the results were presented in Fig. 2. Fig. 2” It is Figure 3(b).

Response 11:

Thank you for your comment. We have carefully checked the manuscript and corrected the errors accordingly.

Point 12:

“As shown in Fig. 3, the absorbance increased when the pH increased from 5 to 7.” Figure 3(c). Actually it is nearly the equal value of pH 6, pH 7, pH 8.

Response 12:

Thank you for your comment. We agree with the reviewer’s assessment. Accordingly, the manuscript, we have revised the contents, "the absorbance increased when the pH increased from 5 to 7. The further increasing pH from 7 to 8 obtained constant absorbance" was corrected to "the absorbance increased when the pH increased from 5 to 6. The further increasing pH from 6 to 8 obtained constant absorbance ".

Point 13:

“The obtained results as 199 shown in Fig. 4”. It is Figure 3(d) “and the results were shown in Fig. 5.” This is Figure 3(e).

Response 13:

Thanks for your careful checks. We are sorry for our carelessness. We have made corrections according to the reviewer’s comments.

Point 14:

Actually, it is not cadmium metal Cd, but cadmium ion (Cd2+) or Cd(II).

Response 14:

Thank you for your comment. We have made corrections according to the reviewer’s comments.

Point 15:

Table 1: Is the values from one single measurement? Or did the authors performed more measurements (n=?). It is kindly suggested to do more than one measurement and to provide the mean and standard deviation or standard error.

Response 15:

Thank you for your comment. We have performed three measurements (n=3) and provided the mean and standard deviation in the revised manuscript and shown below as well.

Point 16:

Table 2: Found +/-SD: no SD data provided. Please add. How about the number of experiments (n=?)

Response 16:

Thank you for your comment. The SD data and the number of experiments (n=3) has been added in the revised manuscript and shown below as well.

Point 17:

How about the values of cadmium in the three different rice samples without added standrads (0 microg/L) regarding toxicity thresholds? Please judge on the obtained values.

Response 17:

Thank you for your comments. We have added the judgement in the revised manuscript and shown below as well.

As shown in Table 2, the results obtained for analysis of these three different rice samples varied from 0.069 to 0.082 µg/g. These results demonstrated that the cadmium concentration was very low in the three rice samples[10].

Point 18:

Table 3: Can the authors also include other works: https://doi.org/10.1016/j.measurement.2019.07.069 from 2019, https://doi.org/10.1007/s13738-011-0018-7, 2017 Journal of Chemical, Biological and Physical Sciences 7(3):578-587? How was the selection of the examples of Table 3 based on by the authors?

Response 19:

Thank you for your introduction to the wonderful research work. According to your suggestion, we properly cite this article as [45] and [46]. 

For comparative purposes, the performance characteristics of the proposed UADLLME method and other preconcentration methods coupled to FAAS.

Point 19:

In conclusion, The authors should stress the advantage of their study compared to the other studies of table 3. What is the advantage of the UADLLME compared to the other extraction styles?

Response 19:

We appreciate for your valuable comment. More description of the advantage of our study has been added in the revised manuscript and shown below as well.

This combination gives good accuracy and performs successfully in the determination of cadmium in rice samples. The UADLLME-FAAS technique with appropriate ligand has several advantages namely low consumption of organic solvent during extraction, high sensitivity, simple and fast operation, good extraction performance lead to short extraction time, low cost and good repeatability. The evaluation of the target compounds yielded satisfactory recovery, linearity precision and detection limit. 

Round 2

Reviewer 1 Report

Comparison studies on several ligands used in determination of Cd (II) in rice by flame atomic absorption spectrometer after ultrasound-assisted dispersive liquid–liquid microextraction

Additional comments after the revision:

Abstract:

  1. Which one should be preferred between “flame atomic absorption spectrometer” and “flame atomic absorption spectrometry”? That one is referred to instrument/apparatus, while another term is referred to “method/technique”.
  2. “Sodium diethyldithiocarbamate (DDTC) both had high distribution coefficient and good absorbance signal,…” It should be better such as “Sodium diethyldithiocarbamate (DDTC) provided high distribution coefficient as well as good absorbance signal, ….”.
  3. “In addition, the experimental conditions of UADLLME were optimized and applied in rice,” Please make sure that the optimization was done in real samples? Or try in std solutions first and then applied to real samples?

Introduction:

  1. L35, Additionally, it is considered to be one of the largest contaminants in photogenic food [2].
  2. L36: “The long-term accumulation of toxins in body…” What is the meaning of toxins? Can we include Cd as a toxin? If not, please revise this statement in the manuscript.
  3. L46-47: Although ICP-OES and ICP-MS are efficient techniques providing sensitivity and selectivity, they are relatively expensive and complicated methods, …. Or the authors can revise to other proper phrases.
  4. L83-94: It seems to be not writing smoothly, because the previous paragraphs concern ligands combined with pre-concentration techniques, and then authors provide the ligands (especially the best ligands for this work) again. I suggest revising again. Alternatively, the authors should re-check the above Refs concerning ligands and different preconcentration methods. If we found that no research works present about the ligands used in this study, we can inform the readers by providing some information to combine the previous one and later paragraphs (as your revised) For example, “However, the ligands containing sulfur atoms (e.g., DDTC, APDC, DMDTC, DBDTC, and DDTP) and N-donor ligands (e.g., PAN, BSE, NTA, EDTA, and DTPA) coupled with the suitable preconcentration methods for Cd have been limited.”
  5. L83-94 (again), please revise to be the concise and short statement. A long description should be placed in the Results and discussion.
  6. Please note that extraction solvents (e.g., heptanol,..) may be less toxic than chlorinated solvents (e.g., Chloroform, carbon tetrachloride, …). So, the authors can include this advantage of the chosen extraction solvent in your manuscript.
  7. Eq (2) should be RR (%)
  8. I wonder why the results concerning RR (%) were almost changed after the revision. This point makes that the reviewer may decide rejection.

Author Response

Reply to Reviewer #1

Abstract:

Point 1:

Which one should be preferred between “flame atomic absorption spectrometer” and “flame atomic absorption spectrometry”? That one is referred to instrument/apparatus, while another term is referred to “method/technique”.

Response 1:

We feel great thanks for your valuable review work on our article. We have preferred “flame atomic absorption spectrometry” and made corrections to our previous draft.    

Point 2:

“Sodium diethyldithiocarbamate (DDTC) both had high distribution coefficient and good absorbance signal,…” It should be better such as “Sodium diethyldithiocarbamate (DDTC) provided high distribution coefficient as well as good absorbance signal, ….”.

Response 2:

Thanks for your very thoughtful suggestion. We have changed "Sodium diethyldithiocarbamate (DDTC) both had high distribution coefficient and good absorbance signal" to "Sodium diethyldithiocarbamate (DDTC) provided high distribution coefficient as well as good absorbance signal".

Point 3:

“In addition, the experimental conditions of UADLLME were optimized and applied in rice,” Please make sure that the optimization was done in real samples? Or try in std solutions first and then applied to real samples?

Response 3:

Thanks for your very thoughtful suggestion. We have changed "In addition, the experimental conditions of UADLLME were optimized and applied in rice" to "In addition, the experimental conditions of UADLLME were optimized in standard solution first and then applied in rice".

Introduction:

Point 4:

L35, Additionally, it is considered to be one of the largest contaminants in photogenic food [2].

Response 4:

Thanks for your very thoughtful suggestion. We have changed "Cadmium is considered to be one of the largest contaminants in phytogenic food" to "Additionally, it is considered to be one of the largest contaminants in phytogenic food".

Point 5:

L36: “The long-term accumulation of toxins in body…” What is the meaning of toxins? Can we include Cd as a toxin? If not, please revise this statement in the manuscript.

Response 5:

Thanks for your suggestion. We have changed "The long-term accumulation of toxins in body" to "The long-term accumulation of cadmium in body".  .

Point 6:

L46-47: Although ICP-OES and ICP-MS are efficient techniques providing sensitivity and selectivity, they are relatively expensive and complicated methods, …. Or the authors can revise to other proper phrases.

Response 6:

Thank you for your comment. We have added the suggested content to the manuscript and shown as below.

Compared with costly analysis technique, such as ICP-OES and ICPMS, FAAS is a conventional technique which has been widely applied in metals analysis because it is a less expensive and simpler alternative technique.

Point 7:

L83-94: It seems to be not writing smoothly, because the previous paragraphs concern ligands combined with pre-concentration techniques, and then authors provide the ligands (especially the best ligands for this work) again. I suggest revising again. Alternatively, the authors should re-check the above Refs concerning ligands and different preconcentration methods. If we found that no research works present about the ligands used in this study, we can inform the readers by providing some information to combine the previous one and later paragraphs (as your revised) For example, “However, the ligands containing sulfur atoms (e.g., DDTC, APDC, DMDTC, DBDTC, and DDTP) and N-donor ligands (e.g., PAN, BSE, NTA, EDTA, and DTPA) coupled with the suitable preconcentration methods for Cd have been limited.”

Response 7:

Thank you for your valuable comment. We have revised the suggested content and shown as below. We hope this time it is smoothly for reading.

The use of complexing agents that selectively reacts with cadmium and subsequent extraction of the obtained metal complex has been widely applied in analysis. For instance, cadmium has been complexed with 8-hydroxyquinoline (8-HQ) [2], 1-(2-thiazolylazo)-p cresol (TAC) [10], ammonium pyrrolidine dithiocarbamate(APDC) [12, 20, 22, 24, 25], 1-(2-pyridylazo)-2-naphthol (PAN) [13, 27], 4,4-dimethyl2,2-bipyridine [15], diphenylcarbazone [16], sodium diethyldithiocarbamate (DDTC) [17, 23, 25], 1,3-dihydroxybenzene [18], L-Cysteine [19], tricaprylmethylammonium chloride(Aliquat® 336) [26], sodium O,O-diethyl dithiophosphate (DEDTP) [33], N,N′-bis(salicylidene)ethylenediamine (BSE) [34], the obtained cadmium complexes extracted by ultrasound-assisted emulsification microextraction with the solidification of floating organic droplet [10], supramolecular solvent based liquid–liquid microextraction [12], solidified floating organic drop microextraction [13], vortex-assisted room temperature ionic liquid dispersive liquid–liquid microextraction [12, 18], green solvent-based ultrasonic assisted dispersive liquid–liquid microextraction [19], switchable hydrophilicity solvent based liquid phase microextraction [20], ultrasound-assisted magnetic retrieval-linked ionic liquid dispersive liquid-liquid microextraction [22], lab-in-syringe magnetic stirring-assisted dispersive liquid-liquid microextraction [24], liquid phase microextraction [28], co-precipitation method [33], magnetic solid-phase extraction [34] and DLLME [2, 15-17, 23, 25-27].

The presence of an appropriate complexing agent for extraction of Cd(II) is necessary in the mentioned DLLMEs[2, 12, 15-19, 22-27], however, there are rare discussions about the effect of types of ligand in DLLMEs. Most of previous researchers demonstrated that the types of extraction solvent and disperser solvent were important in the DLLMEs technique. Actually, the organic ligands play an important role in determination of trace Cd(II), because they are affect not only the extraction efficiency of Cd(II) but also the FAAS behaviour of metals. For example, the extraction conditions of Cd(II) by using DDTC and O,O-diethyl phosphordithiooic acid ammonium(DDTP) are quite different. Acidic conditions are suggested to enhance the extraction efficiency of DDTP, while DDTC is decomposed into diethylamine and carbon disulfide in acidic solution[35]. Even DDTC, APDC and sodium dimethyldithiocarbamate (DMDTC) have similar chemical structures, the stability of the complexes with Cd(II) which will affect the FAAS procedure are different due to their different substituents. Hence, it is important to perform comparative study in order to chose appropriate ligand by consideration of extraction efficiency of Cd(II) as well as atomization degree of Cd(II)-ligand complex.

Point 8:

L83-94 (again), please revise to be the concise and short statement. A long description should be placed in the Results and discussion.

Response 8:

Thank you for your valuable suggestion. We have revised the suggested content and shown as in response 7.

Point 9:

Please note that extraction solvents (e.g., heptanol,..) may be less toxic than chlorinated solvents (e.g., Chloroform, carbon tetrachloride, …). So, the authors can include this advantage of the chosen extraction solvent in your manuscript.

Response 9:

Thank you for your comment. We have added the suggested content in the revised manuscript.

Abstract

A low density and less toxic solvent, 1-heptanol was used as extraction solvent and ethanol was used as disperser solvent.

Results

In previous studies, chlorinated solvents(e.g., chloroform and carbon tetrachloride) in the traditional DLLME method were frequently chosen as the extraction solvent. However, all of those solvents have high toxicity[27]. Therefore, we used the less toxic solvent 1-heptanol and 1-undecanol instead of chlorinated solvents usually used in DLLME.

Conclusions

The UADLLME-FAAS technique with appropriate ligand has several advantages namely low consumption of organic solvent during extraction, less toxic extraction solvent instead of high toxicity chlorinated solvent, high sensitivity, simple and fast operation, good extraction performance lead to short extraction time, low cost and good repeatability.

Point 10:

Eq (2) should be RR (%)

Response 10:

We have corrected it. Thank you for your pointing out.

Point 11:

I wonder why the results concerning RR (%) were almost changed after the revision. This point makes that the reviewer may decide rejection.

Response 11:

Thank you for pointing out this inconsistency. Reviewer 2 kindly suggested us to do more than one measurement and to provide the mean and standard deviation of Table 1 and Table 2. Therefore we performed more measurements and thus the results were changed in Table 1 and Table 2. We are sorry we didn’t mention this in our last response letter.

Reviewer 2 Report

The authors did a diligent job to address all the concerns mentioned before in order to improve the quality of their manuscript. The authors acted on every suggestion and their performance is both reasonable and leads to improvement of their study. Anyways, I wish to mention the following very minor aspects.

@ point 3: The authors Baeker et al. reported on the advantages of AAS compared to ICP-MAS as the following: “Both analytical methods show similar sensitiveness, but the great advantage of AAS over ICP MS are the lower expenses of purchasing and maintaining the spectrometer.47 The simultaneous or sequential measurements further allow multi-element analysis with AAS.47 Moreover, the development of HR CS MAS broadened the applicability to non-metals such as fluorine.48 In contrast, sensitive determination of fluorine employing commercial argon plasma ICP MS is hardly feasible.49,50”

@ Figure 1: The resolution still can be improved. Or was the poor resolution merely caused when creating the pdf-document within the review process? The authors should provide high-resolution images to the MDPI publisher. Moreover, they way how the sulfur anions are presented should be consistent, confer DDTC, DMDTC (actually it is the sulfur and not the sodium bound to the carbon), APDC in contrast to DBDTC, DDTP.

@ Figure 2: there are no error bars included in the figure representing the standard deviation. Please add!

All in all, I suggest further processing of the manuscript in order to publish as an article in MOLECULES after these very minor revision of formal aspects. All the best!

Author Response

Reply to Reviewer #2

Point 1:

@ point 3: The authors Baeker et al. reported on the advantages of AAS compared to ICP-MAS as the following: “Both analytical methods show similar sensitiveness, but the great advantage of AAS over ICP MS are the lower expenses of purchasing and maintaining the spectrometer.47 The simultaneous or sequential measurements further allow multi-element analysis with AAS.47 Moreover, the development of HR CS MAS broadened the applicability to non-metals such as fluorine.48 In contrast, sensitive determination of fluorine employing commercial argon plasma ICP MS is hardly feasible.49,50”

Response 1:

We are extremly sorry for our negligence. Thank you for your introduction to the wonderful research work. According to your suggestion, we properly cite this article 10.1039/c9dt03330k as reference [29].

Point 2:

@ Figure 1: The resolution still can be improved. Or was the poor resolution merely caused when creating the pdf-document within the review process? The authors should provide high-resolution images to the MDPI publisher. Moreover, they way how the sulfur anions are presented should be consistent, confer DDTC, DMDTC (actually it is the sulfur and not the sodium bound to the carbon), APDC in contrast to DBDTC, DDTP.

Response 2:

Thank you for your comment. We have provided Figure 1 with a better resolution   in the revised manuscript and shown below as well. The way how sulfur anions presented are consistent in the revised Figure1.

Figure 1. The chemical structural formulas of ligands.

Point 3:

@ Figure 2: there are no error bars included in the figure representing the standard deviation. Please add!

Response 3:

Thank you for your valuable comments. We have added (the error bars are the standard deviation for n=3) in the caption of Figure 2.

-----End of Reply to Reviewer #2------

We would like to take this opportunity to thank you for all your time involved and this great opportunity for us to improve the manuscript. We hope you will find this revised version satisfactory.

Sincerely,

Wenjuan Lu

Corresponding author: Qian sun, Xinyu Cui, Yanfeng Wang, Pingping Zhang.
